# Novel 3,9-Disubstituted Acridines with Strong Inhibition Activity against Topoisomerase I: Synthesis, Biological Evaluation and Molecular Docking Study

**DOI:** 10.3390/molecules28031308

**Published:** 2023-01-30

**Authors:** Kristína Krochtová, Annamária Halečková, Ladislav Janovec, Michaela Blizniaková, Katarína Kušnírová, Mária Kožurková

**Affiliations:** 1Department of Biochemistry, Faculty of Science, Pavol Jozef Šafárik University in Košice, Moyzesova 11, 040 01 Košice, Slovakia; 2Department of Organic Chemistry, Faculty of Science, Pavol Jozef Šafárik University in Košice, Moyzesova 11, 040 01 Košice, Slovakia

**Keywords:** antiproliferation, acridines, cheminformatics, computational chemistry, nucleic acids, structure–activity relationships, topoisomerase I/IIα

## Abstract

A series of novel 3,9-disubstituted acridines were synthesized and their biological potential was investigated. The synthetic plan consists of eight reaction steps, which produce the final products, derivatives **17a**–**17j**, in a moderate yield. The principles of cheminformatics and computational chemistry were applied in order to study the relationship between the physicochemical properties of the 3,9-disubstituted acridines and their biological activity at a cellular and molecular level. The selected 3,9-disubstituted acridine derivatives were studied in the presence of DNA using spectroscopic (UV-Vis, circular dichroism, and thermal denaturation) and electrophoretic (nuclease activity, relaxation and unwinding assays for topoisomerase I and decatenation assay for topoisomerase IIα) methods. Binding constants (2.81–9.03 × 10^4^ M^−1^) were calculated for the derivatives from the results of the absorption titration spectra. The derivatives were found to have caused the inhibition of both topoisomerase I and topoisomerase IIα. Molecular docking simulations suggested a different way in which the acridines **17a**–**17j** can interact with topoisomerase I versus topoisomerase IIα. A strong correlation between the lipophilicity of the derivatives and their ability to stabilize the intercalation complex was identified for all of the studied agents. Acridines **17a**–**17j** were also subjected to in vitro screening conducted by the Developmental Therapeutic Program of the National Cancer Institute (NCI) against a panel of 60 cancer cell lines. The strongest biological activity was displayed by aniline acridine **17a** (MCF7–GI_50_ 18.6 nM) and *N*,*N*-dimethylaniline acridine **17b** (SR–GI_50_ 38.0 nM). The relationship between the cytostatic activity of the most active substances (derivatives **17a**, **17b**, and **17e**–**17h**) and their values of *K*_B_, Log*P*, Δ*S*°, and δ was also investigated. Due to the fact that a significant correlation was only found in the case of charge density, δ, it is possible to assume that the cytostatic effect might be dependent upon the structural specificity of the acridine derivatives.

## 1. Introduction

Cancer drug research has undergone a remarkable series of changes over the last decade, and recent years have seen many notable successes in the development of cytotoxic drug treatments against cancer. In cancer therapy, DNA acts as a target for two classes of chemotherapeutical: alkylating agents and molecules with non-covalent binding properties. Acridines intercalate between two adjacent base pairs of nucleic acids due to hydrophobic forces and form stable complexes. Proflavine (**1**) and acriflavine (**2**) possess a conjugated positively charged planar structure and have already seen use as DNA-targeted drugs (Figure 1). Acridines can also interact with nucleic acids in ribosomes and polysomes and induce structural dysfunction [1]. Amsacrine (**3**) was the first anticancer agent shown to act by poisoning topoisomerase II [2]. Human DNA encodes two forms of type II topoisomerase (Topo II) isoforms, topoisomerase IIα (Topo IIα) and IIβ (Topo IIβ). The first of these, Topo IIα, is essential for the survival of proliferating cells and increases in concentration over the cell cycle (peaking at G2/M) and is therefore the isoform that is most closely involved in replicative processes [3,4].

Recent studies report on the preparation of new analogues of amsacrine (**3**), and subsequent work has led to the development of the DACA (**4**) and SN 28049 (**5**) derivatives [5,6,7].

Enzyme telomerase is a multisubunit ribonucleoprotein complex that can help to overcome the telomerase shortening problem, which results from the inability of the DNA polymerase to fully replicate the lagging strand during mitosis [8,9]. Trisubstituted acridines of the structure **6** shown in Figure 1 have been synthesized with the aim of targeting telomeric DNA and thereby inducing quadruplex formation [10]. These compounds exhibit higher levels of G-quadruplex affinity than their 3,6-disubstituted analogues [11].

Inubusine B (**7**), the naturally occurring acridine alkaloid, has been isolated from extracts of a culture of *Streptomyces* sp. IFM 11440 [12,13]. Inubusine B (**7**) was recently synthetized and displayed higher levels of induce neurogenesis activity [13].

It was previously reported that 3,9-disubstituted acridine **8** exhibited a high level of antiproliferative activity against prostate cancer cell lines MiaPaCa-2, SU86.86, and BXPC-3 [14]. These findings led to the consideration of the use of 3,9-disubstituted acridines as a new potential source of cytostatic activity. Therefore, we decided to prepare new acridine-based disubstituted compounds taking advantage of some structural features of BRACO-19 (**6**), a molecular pattern with potential anticancer applications.

## 2. Results and Discussion

Spectroscopic techniques (UV-Vis spectroscopy, thermal denaturation, and circular dichroism) were used to study DNA binding properties and to determine the types of DNA interaction with the studied derivatives. The electrophoretic methods were applied to determine the effect of the derivatives on topoisomerase I and IIα. Analyses incorporating cheminformatics and computational chemistry (SAR) were used in the novel series of acridines to determine the relationship between their structure and their activity. Acridines were tested against an NCI panel of 60 cancer cell lines. Docking simulations were used to determine the possible mode of interactions between Topo I/IIα and the novel disubstituted acridines. A schematic view of the experimental methods is presented in Figure 1.

### 2.1. Synthetic Study

As has been noted above, disubstituted acridines display high levels of antiproliferative activity. The aim of this study was to synthesize a series of 3,9-disubstituted acridines to evaluate their biological activity. The first step of this process was to develop a synthetic pathway which could produce the given substances. The synthetic route started with the protection of the amino group of compound **9** followed by the oxidation of toluene **10** with potassium permanganate [15]. The obtained acid **11** was then used in a Jourdan–Ullmann reaction to produce *N*-phenylanthranilic acid **12**. The key step in the overall preparation of the acridines was the cyclisation of anthranilic acid **12** to produce aminoacridone **13**. The formation of acridone **13** proceeded in two steps, firstly the cyclization itself followed by the deprotection of the amino group. The obtained 3-aminoacridone **13** was isolated as a grey solid after the alkalization of the aqueous solution of its reaction mixture. Aminoacridone **13** was then reacted with 2-chloropropionyl chloride to produce compound **14** (Figure 2).

The next step in the process was the preparation of compound **15** through the reaction of derivative **14** with pyrrolidine in ethanol under nitrogen at 55 °C. Product **15** was isolated at a yield of 83%. The next step was the preparation of chloride **16**, one of the key intermediates in the entire synthetic strategy (Figure 3). The yield of the reaction leading to compound **16** did not exceed 60%, probably due to the deprotection of the amino group during the reaction. The final stage in the process was the preparation of the final products, acridines **17a**–**17j**, which were synthesized through the reaction of chloride **16** with their related amines, as is depicted in Figure 3. The purification of the products was performed using two stages of column chromatography on silica gel. The first stage used a mixture of acetone and 26% water solution of diethylamine (30:1–*v/v*) as an eluent to remove crude impurities. The pure products **17a**–**17j** were then isolated by the additional chromatography process using a mixture of ethylacetate and diethylamine in the ratio of 3:1 (*v/v*) as an eluent.

An analysis of proton NMR spectra of crude product **17e** obtained after the first stage of chromatography revealed the presence of an additional side product (Appendix A). Although the sample is contaminated by the main product **17e**, signals that cannot be assigned to the derivative structure **17e** are clearly visible. Another noteworthy finding here is the absence of the proton signals of the pyrrolidine ring. As a next step, 2D NMR methods, including that of ^1^H, ^13^C HMBC, were applied in order to determine the structure of this side product in more detail. Based on the long range ^1^H–^13^C coupling results, the solution would be the structure depicted in Figure 2.

In order to determine the ratio of the main compound **17k** to the side product **17e**, the composition of the related reaction mixture was analyzed using the NMR technique. Based on the integrated intensities of the protons’ signals in the methylene groups of benzyls, the ratio of derivatives **17e** and **17k** was calculated to be approximately 3:1 (Appendix A). The subsequent mass analysis confirmed the hypothesis regarding the structure of the side product, the formation of which was observed during the course of the reaction, which produced compound **17k** (Appendix A).

The formation of the side product **17k** can be explained by the elimination of the pyrrolidine ring leading to the formation of an intermediate (Figure 4). This intermediate, enone **17m**, consequently reacted with benzylamine to produce the side product **17k**. It is important to note here that the elimination can also be promoted by the presence of hydrochloride, which is produced during the course of the reaction. Finally, it should also be noted that the formation of side products was observed in all of the reactions leading to the formation of the final products **17a**–**17j** and not only in the case of compound **17e**, a fact that could explain the moderate yield of the final step in the preparation of derivatives **17a**–**17j**, as is depicted in Figure 2. Although this explanation of the formation of the side products appears to be acceptable in terms of the logic of organic synthesis, the possibility of an alternative reaction mechanism(s) cannot be excluded.

### 2.2. UV-Vis Absorption Spectroscopy

The interaction of a drug with DNA can be corroborated by typical spectral features such as changes in absorbance intensity and a shift in the peak position [16]. Figure 3 and Appendix A and Appendix A show the UV-Vis absorption titration spectra and data for acridine derivatives **17a**–**17i** in both the absence and presence of different concentrations of *ct*DNA.

The interaction of compounds **17a**–**17i** with *ct*DNA were revealed through the presence of hypochromicity (23.30–37.05%) and a slight bathochromic shift (Δλ = 3–7), and the formation of isosbestic points were also observed in the spectra [17]. All of these features indicate that the derivatives have interacted with DNA. Due to the aggregation of one of the studied derivatives **17j** upon the addition of *ct*DNA, only nine of the ten derivatives could be analyzed. The hypochromic effect showed a decrease in the following relation: **17f** > **17e** > **17i** > **17g** > **17h** > **17b** > **17c** > **17d** > **17a**. Using the Benesi–Hildebrand equation, binding constants of 2.81–9.03 × 10^4^ M^−1^ were calculated for the derivatives (Appendix A). Amsacrine, ethidium bromide and Hoechst 33258 were used as reference compounds. While the hypochromic shifts are at the limits of the values that indicate the occurrence of intercalation/groove binding, the presence of bathochromic shifts suggests the likelihood of groove binding. The calculate binding constants were in the middle of the ranges for intercalation and groove binding according to the values of the reference molecules.

### 2.3. Thermal Denaturation

The interaction of small molecules with DNA can influence their melting temperatures (T*m*). The binding of ligands to DNA through intercalation can stabilize the double helical structure of DNA, thereby increasing the T*m* of the ligand–DNA complex by more than 5 °C [18,19]. An investigation of the melting temperature of complexes formed by the studied derivatives and *ct*DNA showed increases in melting temperatures (10–14 °C), which suggests that at least some parts of the derivatives bind to *ct*DNA through intercalation (Figure 4 and Appendix A, Appendix A). The melting temperatures of the complexes formed by derivatives **17h**, **17i** and **17c** could not be determined due to the aggregation of *ct*DNA in the presence of these derivatives under the given experimental conditions. However, due to the similarities between the structures of the studied compounds, we assume that the melting points of these complexes would be within the same range as the values obtained for the other complexes.

### 2.4. Circular Dichroism

Polarized light spectroscopy is another efficient technique for determining and characterizing the interaction of nucleic acids with small molecules. Circular dichroism (CD) is a powerful tool that offers an insight into the structure of these types of complexes [20]. Derivatives **17a**–**17i** are all achiral compounds and therefore do not provide CD spectral profiles. As a result, the changes to the CD spectra of *ct*DNA were observed following the interaction of the *ct*DNA with the studied compounds (Figure 5 and Appendix A). The circular dichroism (ICD) signals observed in the range of 300–500 nm were either non-existent or weak (with the exception of the interaction with derivative **17b**), but the signals in the range of 230–300 nm were much stronger. The interaction of *ct*DNA with the derivatives resulted in an increase in the ellipticity of the positive band and a decrease in the ellipticity of the negative band in the *ct*DNA range. A small batochromic shift and the formation of isoeliptic points were also observed. The presence of isoelliptic points in the spectra suggests the presence of a single binding mechanism [21]. The red shift of the positive band (275 nm) may be the result of the conformational transition of *ct*DNA B-form to A-form through the weakening of the π-π* stacking of base pairs, an effect which may be the result of the insertion of the acridine ring into the DNA [22]. In comparison with the data obtained in the UV-Vis assays, it is possible to conclude that the results of the circular dichroism study support the hypothesis that binding occurred through intercalation in at least one part of the molecule.

### 2.5. Structure–Activity Studies

Analyses incorporating cheminformatics and computational chemistry were applied to the novel acridines in an effort to shed more light on the relationship between their structure and their activity. The first assay studied the relationship between the DNA intercalation capabilities of the synthetized acridines **17a**–**17j** and their physicochemical properties such as log*P*, Δ*S*° and partial charges δ (Table 1). The online cheminformatics portal Online Chemical Database (ochem.eu) was used to obtain the values of lipophilicity, log*P*, while the values of partial charge, δ, and changes of standard entropy, Δ*S*°, were calculated on the semiempirical level of the quantum mechanics theory using the pm7 method [23,24,25,26,27,28,29,30,31]. It should be mentioned here that only a charge on atoms within the substituent at location 9 of the acridine skeleton was used in the QSAR studies (Appendix A). This approach was chosen on the basis that this is the only part of the molecule that changes.

As is shown in Figure 6A, a strong correlation was observed between the lipophilicity of acridines **17a**–**17j** and their ability to stabilize the intercalation complex with *ct*DNA defined in terms of the binding constant *K*_B_. The correlation coefficient, R^2^, of this relationship is nearly 0.9. Previous studies have also addressed the possible relationship between the change of the standard entropy, Δ*S*°, and the binding constant, *K*_B_, in a series of 3,6-disubstituted ureas of acridines [32].

In the case of the 3,9-disubstituted acridines, no such relationship was observed (Figure 6B). In contrast, a strong correlation was observed between entropy changes, Δ*S*°, and binding constants, *K*_B_, for the series of phenylalkyl derivatives **17e**–**17i** (Figure 6B). One interesting point to note here is the fact that the correlation was positive, whereas this was not the case for the 3,6-disubstituted acridines, where the opposite correlation was observed. This would indicate that 3,6 and 3,9 derivatives **17e**, **17h** and **17j** exhibit different modes of binding into the DNA molecule. It is also possible to conclude that the stepwise increase in *K*_B_ values from **17e** to **17i** is probably driven by an increase in enthalpy within the Δ*G*_mol_ term as a consequence of increasing non-bonded interactions between the 9-chains and DNA (Equation (1)) [33,34]. Another possible explanation would be that the increases in lipophilicity in the same manner could give rise to the Δ*G*_hyd_ term (Equation (1)). Yet, another rationalization could arise from a proposed mode of DNA binding where the phenylalkyl chain of acridines **17e**, **17h** and **17i** occupies the minor groove of the DNA helix. This would lead to an expulsion of water molecules forming an inner mesh of water-binding DNA, which would be mirrored in an expansion of binding entropy accommodating the formation of an intercalation complex.
Δ*G*_obs_ = Δ*G*_conf_ + Δ*G*_t+r_ + Δ*G*_hyd_ + Δ*G*_pe_ + Δ*G*_mol_(1)

In the case of a value of partial charges, δ, localized on the 9-substituents, no relationship with the DNA binding constants, *K*_B_, was observed (Appendix A).

Given these results, it is possible to outline some preliminary conclusions. The DNA binding ability of bromo derivative **17c** might be ascribed to its lipophilicity rather than to any structural specificity. In contrast, the ability of phenylalkyl derivatives **17e**, **17h** and **17i** to bind to DNA is strongly dependent on their structure. The rest of the derivatives likely exhibit a combination of these two features that have an impact on DNA complex stabilization, although a measure and an extend could not be specified in simple terms.

### 2.6. Inhibition of Human Topoisomerase I and IIα

Prior to the investigation of human Topo I and Topo IIα inhibition, additional experiments were performed to either verify or exclude the possibility of nuclease activity. The results of these experiments confirmed that none of selected derivatives had triggered the cleavage of pUC19 plasmid (Appendix A). However, a retardation of plasmid mobility was also observed, and this effect was greater at higher concentrations of the studied derivatives, an effect which is a consequence of the binding of selected derivatives with plasmid DNA [35].

### 2.7. Relaxation Assay for Human Topoisomerase I

Once it had been established that none of the novel derivatives possessed the ability to trigger the cleavage of plasmid DNA, it was possible to examine the impact of the compounds on Topo I activity. Firstly, a concentration gradient of Topo I inhibition at increasing concentrations of derivative **17f** was determined in order to identify the appropriate concentration range in which to study the inhibition activity of the derivatives (not shown). According to the results of this initial assay, the suitable concentration range of the derivatives was 1–5 µM, and therefore three concentrations, 1, 2.5 and 5 µM, were adopted for the subsequent assay (Figure 7A). The results of the inhibition assay revealed that the inhibitory effect of the derivatives was reduced as the length of the linker chain between acridine, and benzene ring increased. However, it was also revealed that a substituent at position 4′ on the benzene ring on both the methyl (**17f**) and fluor (**17g**) derivatives caused no significant change in the inhibition activity of these derivatives in comparison to the compound without a substituent on the benzene ring (**17e**). Three concentrations, 1, 5 and 10 µM, were selected for the relaxation assay for samples **17a**–**17d**. The results found that the inhibition activities of the other derivatives were similar to those of the first group, and no significant changes in activity due to the different substituents were identified (Figure 7B).

### 2.8. Unwinding Assay

In order to determine the mechanism of Topo I inhibition, an unwinding assay on supercoiled and relaxed plasmid DNA in the presence of the derivatives was performed. Since the derivatives share a similar skeleton with only small structural changes and a similar inhibition capacity, assays were only performed for one derivative from each of the two groups, compounds **17f** and **17d**, at different concentrations (Figure 8A,B). The results proved that the inhibition of Topo I is caused by the binding of the derivative to the DNA molecule rather than by the inhibition of Topo I itself.

### 2.9. Decatenation Assay for Human Topoisomerase IIα

Catalytic Topo IIα inhibitors are known to act by interfering with the binding between Topo IIα and DNA, by inhibiting ATP binding or by stabilizing the non-covalent DNA-Topo IIα complex. In contrast, Topo IIα poisons can act by stabilizing the covalent DNA-Topo IIα cleavage complex. In order to determine the effect of the studied derivatives, the ability of compounds **17a**–**17i** to affect the decatenation of catenated kinetoplast DNA by Topo IIα was examined (Figure 9A,B). If Topo IIα retained its normal function, the catenated *k*DNA (top band with lowest mobility) would disappear and the bands for open circular intermediate ability and closed circular decatenated *k*DNA (migrating further) would appear. Conversely, the absence of these bands in the results would indicate the inhibition of the enzyme. As is shown in Figure 9A,B, a partial inhibition occurred at a concentration of 10 µM and a stronger inhibitory effect was observed for the derivatives at a concentration of 100 µM. The second group of compounds (**17a**–**17d**) also displayed a weaker inhibition effect on Topo II than derivatives **17e**–**17i**. In comparison with the inhibitory effects for Topo I, it is clear that the studied derivatives show a preference for Topo I over Topo IIα in a ratio approximately than 1:20.

### 2.10. Cytostatic Activity

The synthetized acridines **17a**–**17j** were subjected to in vitro testing against cancer cell lines conducted by the Developmental Therapeutic Program of the National Cancer Institute (NCI). A one-dose screen of each derivative against NCI-60 panels was performed in order to identify the derivatives with the highest biological activity, with acridines **17a**—aniline, **17b**—*N*,*N*-dimethylaniline, **17d**—4-fluorobenzyl, **17e**—benzyl, **17f**—4-methylbenzyl and **17h**—phenylethyl displaying the greatest activity. These derivatives were then evaluated for their growth inhibition activity. From the panel of sixty human cancer cell lines, a single cell line was selected where the derivatives showed the highest cytostatic activity. The results of the evaluation are summarized in Table 2. The most active derivatives are anilines **17a** and **17b** with cytostatic activities in the submicromolar range. The other derivatives in the series discussed above displayed similar levels of activity against the cancer cell lines, as presented in Table 2. In comparison with doxorubicin, the anilines **17a** and **17b** only displayed a higher level of cytostatic activity against the multidrug-resistant ovaria cancer cell line, NCI/ADR-RES (GI_50_[µM]: **Dox**.—10.8 versus **17a**—2.04). We have also compared our results with Paclitaxel, a drug available for the treatment of cancer. Paclitaxel is the first member of the taxane family to be used in cancer chemotherapy. The results are comparable to those with doxorubicin (obtained from the NIH database).

The final point to note is the relationship between the cytostatic activity of the novel substances and their experimental or theoretical physicochemical properties (Table 1). No correlation was observed between cytostatic activity, GI_50_, and the DNA binding ability, *K*_B_, as the correlation coefficients display low values (data not shown). The same conclusion can be drawn in the case of lipophilicity, Log*P*, and the change in the standard enthalpy, Δ*S*°. In contrast, a correlation for the charge density σ was identified within the chains at position 9 of the acridine scaffold. Taking all of these data into consideration, it is possible to suggest that the cytostatic activity of acridines **17a**, **17b**, **17d**–**17f** and **17h** might be a consequence of their molecular structure that selectively affects the enzymatic machinery of the inner cell, which is in good agreement with the strong inhibition activity against Topo I.

### 2.11. Molecular Docking Study

Docking simulations were used to determine the possible mode of interactions between Topo I/IIα and the novel disubstituted acridines **17a**–**17j**. The crystal structure of the ternary DNA cleavage complex for human Topo I (pdb id: 1T8I) was chosen as the receptor for the docking simulations [36]. AutoDockTools-1.5.6 software was used for the preparation of the input files and Autodock ver. 4.2 software was applied to perform the docking of the acridine models [37,38,39,40]. 

The docking simulations for Topo I indicate the possibility of a direct interaction of the positively charged pyrrolidine skeleton of derivatives **17a**–**17j** within a catalytic core consisting of amino acid residues of arginine, lysine, histidine, and 3′-phosphotyrosyl (Figure 10 and Appendix A). Although the docking simulations were performed for the cleavage complex, it cannot be excluded that such interactions can resist the attack of the active site tyrosine on the scissile phosphodiester [41,42].

The docking simulations proposed possible non-bonded hydrophobic interactions between the aniline residue of compound **17a** and Met428, potentially based on electrostatic attractive forces and hydrogen bond formation. Amino acid residue ARG364 together with phosphate group (PO_4_) probably aim to orient the acridine core in the intercalation cavity within the complex with the nucleotide. Electrostatic interactions can be expected between the carboxylic group of ASP533 and the nitrogen atom of the pyrrolidine moiety. The docking simulations also predict the possibility of hydrogen bond formation in the case of NHCO and THR718.

As can be seen in Figure 10, the simulation suggested an intricate mesh of interactions between the pyrrolidinium skeleton of derivative **17a** and the human Topo I catalytic cavity, including ARG488, HIS632, PTR723 (3′-phosphotyrosyl intermediate), and, possibly, LYS532. All of these interactions can lead to the inhibition of human Topo I cleavage activity, more specifically a transesterification reaction where the scissile Np↓N phosphodiester is attacked by a tyrosine of the enzyme, resulting in the formation of a DNA-(3′-phosphotyrosyl)-enzyme intermediate and the expulsion of a 5′-OH polynucleotide. 

Molecular docking studies predicted a similar orientation of derivative **17b** within the ternary DNA cleavage complex for human Topo I (Figure 10). However, the main difference in the case of derivative **17b** is in the orientation of the 4-(dimethylamino)aniline moiety that interacts with the enzyme, with the docking predicted to occur at ASN352 and its close hydrophilic vicinity. A common feature for all derivatives, **17a**–**17j**, is the interaction of the pyrrolidinium ring with amino acid residues that are essential for transesterification. The docking poses of derivatives **17a**–**17j** differ only in the orientation of the aromatic substituents, with the lipophilicity of each compound determining its orientation in the ternary DNA cleavage complex of human Topo I (Appendix A).

In terms of binding energy, there are no significant differences within the studied derivatives **17a**–**17j** (Appendix A), a finding that is in agreement with the results of experimental observations in which no single derivative showed a substantially higher level of inhibition activity. On the other hand, the theoretical molecular docking simulation proposes the likelihood of the direct interaction of the acridine substances **17a**–**17j** with the binding pocket of human Topo I, which reflects their high inhibition potential.

We used the same theoretical approach to determine possible interactions between the novel acridines **17a**–**17j** and human Topo IIα at the molecular level [43]. For this purpose, we selected the only published crystal structure of human Topo IIα in complex with etoposide as a receptor for use in the docking simulations. Unfortunately, the results of these simulations appeared to be logically inconsistent and could not be accepted as accurate (data not shown). These misleading results may be attributed to the structural differences between the intercalating acridines **17a**–**17j** and the non-intercalating etoposide, whose ternary DNA cleavage complex with Topo IIα does not appear to be suitable for use in acridine docking studies.

Given this drawback, we were forced to compromise and use a complex of amsacrine and Topo IIβ for docking simulations [44]. Our decision was based on the fact that Topo IIα and IIβ are isoforms whose binding site residues display a high degree of homology, with a 91.4% sequence identity match. In fact, only five binding pocket residues are not shared by the two isoforms: Thr468/Ser483, Met762/Gln778, Ser763/Ala779, Ile769/Val785 and Ser800/Ala816 (isoform α/β) [45].

The docking simulations predicted the possibility of non-bonding interactions of the docked acridine models **17a**–**17b**, with the nucleotide acting as the main binding force that can stabilize the complex with the enzyme. Figure 11 illustrates the interactions of the phosphodiester backbone of the nucleotide with pyrrolidinium and amide moieties via hydrogen bond formations. Very similar non-bonding interactions accompanying the complex formation can be observed for all of the acridine derivatives, **17a**–**17j** (Appendix A). Further stabilization may result from the attractive non-covalent interactions of the molecular surfaces of 9-aniline/phenylalkyl substituents and the amino acid residues around ARG503.

In the terms of binding energy, no substantial differences were observed between acridines **17a**–**17j**, a finding which correlates with experimental observations where the substances exhibited a similar level of anti-enzymatic activity against human Topo IIβ(α) (Appendix A).

It should be emphasized here that the molecular docking simulations suggest that there is a difference between the way in which acridines **17a**–**17j** interact with Topo I and the way in which they interact with Topo II IIβ(α). In the case of Topo I, the acridines directly attack the binding pocket residues, while non-bonded interactions with the nucleotide backbone were more typical in the case of the cleavage complex for Topo II IIβ(α). This conclusion is in agreement with the experimental results, which clearly demonstrate that the inhibitory activity of acridines **17a**–**17j** against human Topo I is approximately 20-times higher than that against human Topo IIα. Overall, 3,9-disubstituted acridines **17a**–**17j** appear to offer a suitable molecular pattern for the potential next generation of acridine-based drugs with antiproliferative activity due to their potent inhibitory activity against human Topo I.

## 3. Materials and Methods

### 3.1. Chemistry

All reagents were purchased from commercial suppliers without further purification. Reactions were monitored using Thin Layer Chromatography (TLC) on plates (GF**_254_**) supplied by Merck, visualized with a UV lamp. Column chromatography was performed using E. Merck silica gel (60, particle size 0.040–0.063 mm). NMR spectra were recorded on Varian VNMRS NMR (^1^H—600 MHz, ^13^C—150 MHz) and Varian Mercury Plus 400 FT NMR (^1^H—400 MHz, ^13^C—100 MHz) spectrometers at room temperature. Chemical shifts (δ) for ^1^H and ^13^C{^1^H} NMR are given in parts per million (ppm) using the residual solvent peaks as a reference relative to tetramethylsilane (0.00 ppm). Coupling constant (J) values are reported in Hz. High-resolution mass spectra (HRMS) were recorded on a micrOTOF-QII quadrupole-time-of-flight mass spectrometer (Bruker Daltonics) with an electrospray ionization source. Melting points were recorded on a Kofler hot block and are presented uncorrected. Ethidium bromide, dimethyl sulfoxide, amsacrine, Hoechst 33258, calf thymus DNA, tris(hydroxymethyl) aminomethane, campthotecine and agarose were obtained from Sigma-Aldrich.

Synthesis of 3-{[3-(pyrolidin-1-yl)propanoyl]amino}-9-substituted acridines **17a**–**17j**: A mixture of 3-{[3-(pyrrolidin-1-yl)propanoyl]amino}-9-chloroacridine (**16**) (0.1 g, 0.3 mmol) and an appropriate amine (0.4 mmol) in DMF (1 mL) was heated at 100 °C in an oil bath for two hours. The reaction was monitored using TLC chromatography (eluent-MeOH/26% NH_4_OH 30:1 by volume). The reaction mixture was then cooled and diluted with 20 mL of methanol. The resultant mixture was poured into 200 mL of water and the product was precipitated by the dropwise addition of concentrated sodium hydroxide to an alkaline pH level. The heterogeneous mixture was allowed to stand for 24 h at room temperature and then the product was filtered off. Prior to the final purification, all crude products were eluted using a mixture of acetone and 26% water solution of diethylamine (30:1/v:v) on a column of silica gel, except benzylamine derivatives **17d**, **17e**, **17f**, where this elution was sufficient. All products were then purified by column chromatography on silica gel using a mixture of ethyl acetate–diethylamine in a ratio of 3:1. All products were crystalized from ethylacetate with the exception of aniline derivatives **17a**–**17d** that were crystalized from acetone. Aniline derivatives **17a**–**17d** were converted to their hydrochloride salts and their NMR spectra were measured. Melting points and MS were also measured for aniline derivatives **17a**–**17d**. The compound **17k** was isolated as the side product within the synthesis of the substance **17e**.

*N-[9-(Phenylamino)acridin-3-yl]-3-(pyrrolidin-1-yl)propanamide* (**17a**). Yellow crystalline solid; weight 0.04 g; yield 34%; *R*_f_ = 0.62 (Et_2_NH/Ethylacetate 1:3); m.p. 218–200 °C; ^1^H NMR (600 MHz, Metanol–*d*_4_): δ = 8.61–8.54 (m, 1H; H4), 8.19 (d, *J* = 8.7 Hz, 1H; H8), 8.00–7.89 (m, 2H; H1, H6), 7.90–7.83 (m, 1H; H5), 7.58–7.48 (m, 2H; H3′, H5′), 7.48–7.43 (m, 1H; H4′), 7.43–7.39 (m, 3H; H2′, H6′, H7), 7.36–7.26 (m, 1H; H2), 3.80–3.70 (m, 2H; Npy(CH_2A_CH_2_)), 3.63 (t, *J* = 6.8 Hz, 2H; COCH_2_CH_2_), 3.25–3.15 (m, 2H; Npy(CH_2B_CH_2_)), 3.09 (t, *J* = 6.8 Hz, 2H; COCH_2_CH_2_), 2.25–2.15 (m, 2H; Npy(CH_2_CH_2A_)), 2.13–2.03 ppm (m, 2H; Npy(CH_2_CH_2B_)); ^13^C NMR (150 MHz, Metanol–*d*_4_): δ = 170.8 (CO), 156.2 (C9), 146.1 (C4a), 143.6 (C3), 142.2 (C1′), 141.6 (C10a), 136.5 (C6), 131.4 (C3′,C5′), 129.0 (C4′), 128.2 (C1), 126.5 (C8), 126.0 (C2′,C6′), 125.3 (C7), 119.9 (C5), 118.5 (C2), 115.3 (C8a), 111.2 (C9a), 106.5 (C4), 55.5 (Npy(CH_2_CH_2_)_2_), 51.7 (COCH_2_CH_2_), 33.6 (COCH_2_CH_2_), 24.1 ppm (Npy(CH_2_CH_2_)_2_); HRMS (ESI): *m/z* calcd for C_26_H_26_N_4_O+H^+^: 411.21794 [M + H]^+^; found 411.21894.

*N-(9-{[4-(Dimethylamino)phenyl]amino}acridin-3-yl)-3-(pyrrolidin-1-yl)propanamide* (**17b**). Brown crystalline solid; weight 0.02 g; yield 15%; *R*_f_ = 0.40 (Et_2_NH/Ethylacetate 1:3); m.p. 198–200 °C; ^1^H NMR (600 MHz, Metanol–*d*_4_/D_2_O–5:1): δ = 8.60 (d, H4, 1H; *J* = 2.2 Hz), 8.25–8.22 (m, 1H; H8), 8.09 (d, *J* = 9.4 Hz, 1H; H1), 8.02–7.98 (m, 1H; H6), 7.96–7.93 (m, 1H; H5), 7.52–7.47 (m, 1H; H7), 7.46–7.42 (m, 4H; H2′, H3′, H5′, H6′), 7.40 (dd, *J* = 9.4, 2.2 Hz, 1H; H2), 3.78–3.71 (m, 2H; Npy(CH_2A_CH_2_)), 3.62 (t, *J* = 6.8 Hz, 2H; COCH_2_CH_2_), 3.23 (s, 6H; (CH_3_)_2_), 3.22–3.15 (m, 2H; Npy(CH_2B_CH_2_)), 3.09 (t, *J* = 6.8 Hz, 2H; COCH_2_CH_2_), 2.24–2.18 (m, 2H; Npy(CH_2_CH_2A_)), 2.12–2.05 ppm (m, 2H; Npy(CH_2_CH_2B_)); ^13^C NMR (150 MHz, Metanol–*d*_4_*/*D_2_O–5:1): δ = 171.2 (CO), 155.8 (C9), 145.8 (C4a), 145.6 (C4′), 143.4 (C3), 141.6 (C10a), 138.8 (C1′), 136.8 (C6), 128.1 (C1), {126.5(120.0) (C2′,C3′,C5′,C6′,C5)}, 126.4 (C8), 125.7 (C7), 119.0 (C2), 115.6 (C8a), 111.9 (C9a), 106.8 (C4), 55.6 (Npy(CH_2_CH_2_)_2_), 51.5 (COCH_2_CH_2_), 44.8 (CH_3_), 33.6 (COCH_2_CH_2_), 24.0 ppm (Npy(CH_2_CH_2_)_2_). HRMS (ESI): *m/z* calcd for C_28_H_31_N_5_O+H^+^: 454.26014 [M + H]^+^; found 454.26098.

*N-{9-[(4-Bromophenyl)amino]acridin-3-yl}-3-(pyrrolidin-1-yl)propanamide* (**17c**). Yellow crystalline solid; weight 0.06 g; yield 43%; *R*_f_ = 0.52 (Et_2_NH/Ethylacetate 1:3); m.p. 223–225 °C; ^1^H NMR (600 MHz, Metanol–*d*_4_): δ = 8.68 (d, *J* = 2.2 Hz, 1H; H4), 8.25 (d, *J* = 8.6 Hz, 1H; H8), 8.06 (d, *J* = 9.4 Hz, 1H; H1), 8.01–7.96 (m, 1H; H6), 7.93–7.90 (m, 1H; H5), 7.69–7.65 (m, 2H; H3′, H5′), 7.52–7.49 (m, 1H; H7), 7.40 (dd, *J* = 9.4, 2.2 Hz, 1H; H2), 7.35–7.32 (m, 2H; H2′, H6′), 3.76–3.70 (m, 2H; Npy(CH_2A_CH_2_)), 3.63 (t, *J* = 6.8 Hz, 2H; COCH_2_CH_2_), 3.23–3.16 (m, 2H; Npy(CH_2B_CH_2_)), 3.08 (t, *J* = 6.8 Hz, 2H; COCH_2_CH_2_), 2.24–2.16 (m, 2H; Npy(CH_2_CH_2A_)), 2.10–2.03 ppm (m, 2H; Npy(CH_2_CH_2B_)); ^13^C NMR (150 MHz, Metanol–*d*_4_): δ = 170.9 (CO), 156.1 (C9), 146.3 (C4a), 143.8 (C3), 141.8 (C1′), 141.7 (C10a), 136.7 (C6), 134.4 (C3′,C5′), 128.2 (C1), 127.4 (C2′,C6′), 126.5 (C8), 125.6 (C7), 121.8 (C4′), 120.0 (C5), 118.9 (C2), 115.7 (C8a), 111.7 (C9a), 106.6 (C4), 55.6 (Npy(CH_2_CH_2_)_2_), 51.7 (COCH_2_CH_2_), 33.5 (COCH_2_CH_2_), 24.1 ppm (Npy(CH_2_CH_2_)_2_). HRMS (ESI): *m/z* calcd for C_26_H_25_BrN_4_O 489.12845+H^+^: [M + H]^+^; found 489.13023.

*N-{9-[(4-Fluorophenyl)amino]acridin-3-yl}-3-(pyrrolidin-1-yl)propanamide* (**17d**). Yellow crystalline solid; weight 0.05 g; yield 41%; *R*_f_ = 0.63 (Et_2_NH/Ethylacetate 1:3); m.p. 223–225 °C; ^1^H NMR (600 MHz, Metanol–*d*_4_): δ = 8.62 (d, *J* = 2.2 Hz, 1H; H4), 8.20 (d, *J* = 8.6 Hz, 1H; H8), 8.02–7.93 (m, H1, H6, 2H), 7.90–7.86 (m, 1H; H5), 7.49–7.42 (m, 3H; H3′, H5′, H7), 7.36 (dd, *J* = 9.4, 2.2 Hz, 1H; H2), 7.31–7.25 (m, 2H; H2′, H6′), 3.77–3.71 (m, 2H; Npy(CH_2A_CH_2_)), 3.63 (t, *J* = 6.8 Hz, 2H; COCH_2_CH_2_), 3.24–3.17 (m, 2H; Npy(CH_2B_CH_2_)), 3.09 (t, *J* = 6.8 Hz, 2H; COCH_2_CH_2_), 2.23–2.16 (m, 2H; Npy(CH_2_CH_2A_)), 2.10–2.04 ppm (m, 2H; Npy(CH_2_CH_2B_)); ^13^C NMR (150 MHz, Metanol–*d*_4_): δ = 170.9 (CO), 163.2 (d, *J* = 247.1 Hz; C4′), 156.4 (C9), 146.1 (C4a), 143.6 (C3), 141.7 (C10a), 138.3 (C1′), 136.6 (C6), 128.2 (d, *J* = 8.6 Hz; C3′,C5′), 128.1 (C1), 126.4 (C8), 125.3 (C7), 119.9 (C5), 118.6 (C2), 118.2 (d, *J* = 23.3 Hz; C2′,C6′), 115.2 (C8a), 111.1 (C9a), 106.6 (C4), 55.6 (Npy(CH_2_CH_2_)_2_), 51.7 (COCH_2_CH_2_), 33.6 (COCH_2_CH_2_), 24.1 ppm (Npy(CH_2_CH_2_)_2_). HRMS (ESI): *m/z* calcd for C_26_H_25_FN_4_O+H^+^: 429.20852 [M + H]^+^; found 429.20920.

*N-[9-(Benzylamino)acridin-3-yl]-3-(pyrrolidin-1-yl)propanamide* (**17e**). Yellow crystalline solid; weight 0.03 g; yield 25%; *R*_f_ = 0.51 (Et_2_NH/Ethylacetate 1:3); m.p. 150–152 °C; ^1^H NMR (600 MHz, Chloroform–*d*): δ = 11.65 (s, 1H; CONH), 8.07–7.97 (m, 3H; H1, H5, H8), 7.89 (d, *J =* 9.4 Hz, 1H; H2), 7.83 (d, *J =* 2.1 Hz, 1H;), 7.63 (dd, *J =* 8.6, 6.7 Hz, 1H; H6), 7.41–7.36 (m, 4H; H3′, H5′, H2′, H6′), 7.35–7.31 (m, H4; H4′, 1H), 7.31–7.27 (m, 1H; H7), 4.95 (s, 2H; NHCH_2_), 2.87 (t, *J* = 7.1 Hz, 2H; COCH_2_CH_2_), 2.74–2.66 (m, 4H; N_py_CH_2_CH_2_), 2.57 (t, *J* = 7.1 Hz, 2H; COCH_2_CH_2_), 1.92 ppm (m, 4H; N_py_CH_2_CH_2_). ^13^C NMR (150 MHz, Chloroform–*d*): δ = 151.3 (C9), 150.0 (C10a), 149.5 (C4a), 140.9 (C3), 139.4 (C1′), 130.2 (C6), 129.2 (C3′,C5′), 128.1 (C4′), 127.7 (C2′,C6′), 124.3 (C5), 122.9 (C1,C7,C8), 118.5 (C2), 116.7 (C8a), 115.2 (C4), 113.9 (C10a), 55.1 (NHCH_2_), 53.3 (N_py_CH_2_CH_2_), 51.5 (COCH_2_CH_2_), 34.9 (COCH_2_CH_2_), 23.9 ppm (N_py_CH_2_CH_2_). HRMS (ESI): *m/z* calcd for C_27_H_28_N_4_O+H^+^: 425.23359 [M + H]^+^; found 425.23487.

*N-(9-{[(4-Methylphenyl)methyl]amino}acridin-3-yl)-3-(pyrrolidin-1-yl)propanamide* (**17f**). Yellow crystalline solid; weight 0.04 g; yield 32%; *R*_f_ = 0.45 (Et_2_NH/Ethylacetate 1:3); m.p. 145–147 °C; ^1^H NMR (600 MHz, Metanol–*d*_4_): δ = 8.24 (d, *J* = 2.2 Hz, 1H; H4), 8.20 (d, *J* = 8.1 Hz, 1H; H8), 8.16 (d, *J* = 9.4 Hz, 1H; H1), 7.84–7.81 (m, 1H; H5), 7.65–7.61 (m, 1H; H6), 7.38 (dd, *J* = 9.4, 2.2 Hz, 1H; H2), 7.27–7.24 (m, 3H; H7, H2′, H6′), 7.15–7.12 (m, 2H; H3′, H5′), 4.98 (s, 2H; CH_2_), 2.89 (t, *J* = 7.3 Hz, 2H; COCH_2_CH_2_), 2.68–2.59 (m, 6H; COCH_2_CH_2_, Npy(CH_2_CH_2_)_2_), 2.30 (s, 3H; CH_3_), 1.87–1.81 ppm (m, 4H; Npy(CH_2_CH_2_)_2_); ^13^C NMR (150 MHz, Metanol–*d*_4_): δ = 173.1 (CO), 154,4 (C9), 150.3 (C4a), 149.5 (C10a), 142.2 (C3), 138.3 (C4′), 137.6 (C1′), 131.9 (C6), 130.4 (C3′,C5′), 128.0 (C2′,C6′), 127.3 (C5), 126.3 (C1), 124.9 (C8), 123.4 (C7), 118.0 (C2), 117.0 (C8a), 114.6 (C4), 113.7 (C9a), 54.8 (Npy(CH_2_CH_2_)_2_), 53.9 (CH_2_), 52.8 (COCH_2_CH_2_), 36.9 (COCH_2_CH_2_), 24.3 (Npy(CH_2_CH_2_)_2_), 21.1 ppm (CH_3_); HRMS (ESI): *m/z* calcd for C_28_H_30_N_4_O+H^+^: 439.24924 [M + H]^+^; found 439.25051.

*N-(9-{[(4-Fluorophenyl)methyl]amino}acridin-3-yl)-3-(pyrrolidin-1-yl)propanamide* (**17g**). Yellow crystalline solid; weight 0.04 g; yield 32%; R_f_ = 0.54 (Et_2_NH/Ethylacetate 1:3) m.p. 101–103 °C; ^1^H NMR (600 MHz, Chloroform–*d*): δ = 11.68 (s, 1H; CONH), 8.06–7.94 (m, 3H; H1, H5, H8), 7.92–7.80 (m, 2H; H2, H4), 7.67–7.59 (m, 1H; H6), 7.37–7.32 (m, 2H; H2′, H6′), 7.32–7.27 (m, 1H; H7), 7.08–7.02 (m, 2H; H3′, H5′), 4.91 (s, 2H; CH_2_), 2.87 (t, J = 6.6 Hz, 2H; COCH_2_CH_2_), 2.74–2.66 (m, 4H; Npy(CH_2_CH_2_)_2_), 2.57 (t, J = 6.6 Hz, 2H; COCH_2_CH_2_), 1.97–1.88 ppm (m, 4H; Npy(CH_2_CH_2_)_2_); ^13^C NMR (150 MHz, Chloroform–*d*): δ = 171.6 (CO), 162.5 (d, J = 246.5 Hz; C4′), 151.1 (C9), 149.5 (C4a,C10a), 140.9 (C3), 135.2 (C1′), 130.2 (C6), 129.4 (d, J = 8.1 Hz; C2′,C6′), 124.3 (C5), 123.0 (C7), 122.9 (C1,C8), 118.6 (C2), 117.0 (C8a), 116.1 (d, J = 21.5 Hz; C3′,C5′), 115.1 (C4), 114.1 (C9a), 54.4 (CH_2_), 53.3 (Npy(CH_2_CH_2_)_2_), 51.5 (COCH_2_CH_2_), 34.9 (COCH_2_CH_2_), 23.9 ppm (Npy(CH_2_CH_2_)_2_). HRMS (ESI): m/z calcd for C_27_H_27_FN_4_O+H^+^: 443.22417 [M + H]^+^; found 443.22616.

*N-(9-{[(2-Phenylethyl)]amino}acridin-3-yl)-3-(pyrrolidin-1-yl)propanamide* (**17h**). Yellow crystalline solid; weight 0.05 g; yield 40%; *R*_f_ = 0.63 (Et_2_NH/Ethylacetate 1:3); m.p. 110–112 °C; ^1^H NMR (600 MHz, Metanol–*d*_4_): δ = 8.22 (d, *J* = 2.2 Hz, 1H; H4), 8.15–8.11 (m, 2H; H1, H8), 7.82 (d, *J* = 8.7 Hz, 1H; H5), 7.64–7.60 (m, 1H; H6), 7.46 (dd, *J* = 9.4, 2.2 Hz, 1H; H2), 7.30–7.25 (m, 1H; H7), 7.22–7.15 (m, 4H; H2′, H3′, H5′, H6′), 7.15–7.10 (m, 1H; H4′), 4.08 (t, *J* = 7.9 Hz, 2H; NHCH_2_CH_2_Ph), 3.04 (t, *J* = 7.3 Hz, 2H; NHCH_2_CH_2_Ph), 2.89 (t, *J* = 7.3 Hz, 2H; COCH_2_CH_2_), 2.70–2.58 (m, 6H; COCH_2_CH_2_, Npy(CH_2_CH_2_)_2_), 1.87–1.81 ppm (m, 4H; Npy(CH_2_CH_2_)_2_); ^13^C NMR (150 MHz, Metanol–*d*_4_): δ = 173.1 (CO), 154.0 (C9), 150.5 (C4a), 149.8 (C10a), 142.0 (C3), 139.9 (C1′), 131.7 (C6), 129.9 (C2′,C6′), 129.6 (C3′,C5′), 127.6 (C5), 127.5 (C4′), 126.0 (C1), 124.7 (C8), 123.3 (C7), 118.1 (C2), 117.1 (C8a), 115.0 (C4), 114.0 (C9a), 54.8 (Npy(CH_2_CH_2_)_2_), 52.8 (COCH_2_CH_2_), 52.4 (NHCH_2_CH_2_Ph), 38.0 (NHCH_2_CH_2_Ph), 36.9 (COCH_2_CH_2_), 24.3 ppm (Npy(CH_2_CH_2_)_2_). HRMS (ESI): *m/z* calcd for C_28_H_30_N_4_O+H^+^: 439.24924 [M + H]^+^; found 439.25038.

*N-(9-{[(3-Phenylpropyl)]amino}acridin-3-yl)-3-(pyrrolidin-1-yl)propanamide* (**17i**). Yellow crystalline solid; weight 0.05 g; yield 39%; *R*_f_ = 0.48 (Et_2_NH/Ethylacetate 1:3); m.p. 79–81 °C; ^1^H NMR (600 MHz, Metanol–*d*_4_): δ = 8.22 (d, *J* = 2.2 Hz, 1H; H4), 8.15 (d, *J* = 8.1 Hz, 1H; H8), 8.11 (d, *J* = 9.4 Hz, 1H; H1), 7.82 (d, *J* = 8.7 Hz, 1H; H5), 7.64–7.60 (m, 1H; H6), 7.42 (dd, *J* = 9.4, 2.2 Hz, 1H; H2), 7.28–7.23 (m, 1H; H7), 7.20–7.15 (m, 2H; H3′, H5′), 7.13–7.08 (m, 1H; H4′), 7.08–7.04 (m, 2H; H2′, H6′), 3.83 (t, *J* = 7.1 Hz, 2H; NHCH_2_CH_2_CH_2_Ph), 2.90 (t, *J* = 7.3 Hz, 2H; COCH_2_CH_2_), 2.69–2.64 (m, 4H; COCH_2_CH_2_, NHCH_2_CH_2_CH_2_Ph), 2.64–2.61 (m, 4H; Npy(CH_2_CH_2_)_2_), 2.14–2.07 (m, 2H; NHCH_2_CH_2_CH_2_Ph), 1.88–1.81 ppm (m, 4H; Npy(CH_2_CH_2_)_2_); ^13^C NMR (150 MHz, Metanol–*d*_4_): δ = 173.1 (CO), 154.2 (C9), 150.5 (C4a), 149.7 (C10a), 142.5 (C1′), 142.1 (C3), 131.8 (C6), 129.5 (C2′,C6′), 129.4 (C3′,C5′), 127.4 (C5), 127.0 (C4′), 126.1 (C1), 124.7 (C8), 123.2 (C7), 117.9 (C2), 116.9 (C8a), 114.9 (C4), 113.6 (C9a), 54.8 (Npy(CH_2_CH_2_)_2_), 52.8 (COCH_2_CH_2_), 50.3 (NHCH_2_CH_2_CH_2_Ph), 36.9 (COCH_2_CH_2_), 34.0 (NHCH_2_CH_2_CH_2_Ph), 33.6 (NHCH_2_CH_2_CH_2_Ph), 24.3 ppm (Npy(CH_2_CH_2_)_2_). HRMS (ESI): *m/z* calcd for C_29_H_32_N_4_O+H^+^: 453.26489 [M + H]^+^; found 453.26610.

*N-(9-{[(4-Phenylbutyl)]amino}acridin-3-yl)-3-(pyrrolidin-1-yl)propanamide* (**17j**). Yellow crystalline solid; weight 0.06 g; yield 38%; *R*_f_ = 0.62 (Et_2_NH/Ethylacetate 1:3); m.p. 65–67 °C; ^1^H NMR (600 MHz, Metanol–*d*_4_): δ = 8.26–8.20 (m, 3H; H1, H4, H8), 7.84 (d, *J* = 8.7 Hz, 1H; H5), 7.68–7.64 (m, 1H; H6), 7.50 (dd, *J* = 9.4, 2.2 Hz, 1H; H2), 7.35–7.30 (m, 1H; H7), 7.20–7.15 (m, 2H; H3′, H5′), 7.12–7.07 (m, 1H; H4′), 7.07–7.04 (m, 2H; H2′, H6′), 3.88 (t, *J* = 7.1 Hz, 2H; NHCH_2_CH_2_CH_2_CH_2_Ph), 2.92 (t, *J* = 7.3 Hz, 2H; COCH_2_CH_2_), 2.71–2.62 (m, 6H; COCH_2_CH_2_, Npy(CH_2_CH_2_)_2_), 2.58 (t, *J* = 7.5 Hz, 2H; NHCH_2_CH_2_CH_2_CH_2_Ph), 1.90–1.83 (m, 4H; Npy(CH_2_CH_2_)_2_), 1.84–1.78 (m, 2H; NHCH_2_CH_2_CH_2_CH_2_Ph), 1.73–1.66 ppm (m, 2H; NHCH_2_CH_2_CH_2_CH_2_Ph). ^13^C NMR (150 MHz, Metanol–*d*_4_): δ = 173.1 (CO), 154.5 (C9), 150.4 (C4a), 149.6 (C10a), 143.3 (C1′), 142.2 (C3), 131.9 (C6), 129.3 (C2′,C6′,C3′,C5′), 127.3 (C5), 126.8 (C4′), 126.2 (C1), 124.8 (C8), 123.4 (C7), 118.0 (C2), 116.9 (C8a), 114.7 (C4), 113.7 (C9a), 54.8 (Npy(CH_2_CH_2_)_2_), 52.8 (COCH_2_CH_2_), 50.8 (NHCH_2_CH_2_CH_2_CH_2_Ph), 36.9 (COCH_2_CH_2_), 36.3 (NHCH_2_CH_2_CH_2_CH_2_Ph), 31.4 (NHCH_2_CH_2_CH_2_CH_2_Ph), 29.7 (NHCH_2_CH_2_CH_2_CH_2_Ph), 24.3 ppm (Npy(CH_2_CH_2_)_2_). HRMS (ESI): *m/z* calcd for C_30_H_34_N_4_O+H^+^: 467.28054 [M + H]^+^; found 467.28187.

### 3.2. DNA Binding Experiments

Selected derivatives were divided into two testing groups, the first of which consisted of derivatives **17e**–**17i** and the second of derivatives **17a**–**17d**. The absorption spectra of the free acridine derivatives and their complexes with *ct*DNA were measured using a Specord S300 UV-Vis spectrophotometer in a 10 mM Tris-HCl buffer (pH 7.4) at room temperature in a 100-QS quartz cuvette (1 cm path length). The measured UV-Vis data for derivatives **17a**–**17i** were processed graphically using GraphPad Prism 6 software. Binding constants K_B_ were calculated using the Benesi–Hildebrand equation. (Appendix A). *Tm* measurements were carried out on a Jasco J-810 spectropolarimeter in a 1 mm quartz cuvette 100-QS in a BPES buffer (6 mM Na_2_HPO_4_, 2 mM NaH_2_PO_4_, 1 mM Na_2_EDTA, 35 mM NaCl, pH 7.1). Absorption thermal denaturation curves were processed using GraphPad Prism 6, and the results were used to determine the melting temperatures. The CD spectra measurements were conducted using a J-810 Jasco spectropolarimeter in a 100-QS quartz cuvette (1 cm path length) in a 10 mM Tris-HCl buffer (pH 7.4) at room temperature. The results are presented as the mean of at least three independent measurements, and the obtained data were processed using GraphPad Prism 6.

### 3.3. hTopoisomerase I and IIα Experiments

The nuclease activity of selected molecules was studied on isolated plasmid pUC19 according to the adjusted standard protocol. The impact of the studied derivatives on the relaxation ability of *h*Topo I (Inspiralis) was studied with plasmid pBR322 (Inspiralis).

*h*Topo I relaxation assays were carried out according to the Inspiralis protocol with supercoiled pBR322 plasmid (0.5 µg per sample) in a 1 × concentrated assay buffer (20 mM Tris-HCl (pH 7.5), 200 mM NaCl, 0.25 mM EDTA, 5% *v/v* glycerol, 50 µg/mL albumin, supplied as 10 × stock) and an appropriate amount of diluted *h*Topo I (0.8 U per sample (**17e**–**i**) and 1.0 U per sample (**17a**–**d**)) in a dilution buffer (10 mM Tris-HCl (pH 7.5), 1 mM DTT, 1 mM EDTA, 50% *v/v* glycerol, 50 μg/mL albumin).

*wg*Topo I unwinding assays were carried out according to the Inspiralis protocol with supercoiled and relaxed pBR322 plasmid (0.3 µg per sample) in a 1 × concentrated assay buffer (50 mM Tris-HCl (pH 7.9), 50 mM NaCl, 1 mM EDTA, 1 mM DTT, 20% *v/v* glycerol, supplied as 2 × stock) and an appropriate amount of diluted wheat germ *wg*Topo I (0.8 U per sample) in a dilution buffer (50 mM Tris-HCl (pH 7.9), 500 mM NaCl, 1 mM DTT, 1 mM EDTA, 50% *v/v* glycerol).

*h*Topo IIα decatenation assays were carried out according to the Inspiralis protocol using kinetoplast DNA (*k*DNA, 200 ng per sample) in a 1 × concentrated assay buffer (50 mM Tris-HCl (pH 7.5), 125 mM NaCl, 10 mM MgCl_2_, 5 mM DTT, 100 µg/mL albumin, supplied as 10 × stock) with ATP (final concentration 1 mM) and an appropriate amount of diluted *h*Topo IIα enzyme (1.2 U per sample) in a dilution buffer (50 mM Tris-HCl (pH 7.5), 100 mM NaCl, 1 mM DTT, 0.5 mM EDTA, 50% *v/v* glycerol, 50 μg/mL albumin).

### 3.4. Screening of Anticancer Activity-NCI-60 Panels

The anticancer activity of acridines **17a**–**17j** was tested against NCI-60 panels consisting of sixty human cancer cell lines. The data about NCI-60 human cancer cell lines are available at https://dtp.cancer.gov/discovery_development/nci-60/cell_list.htm (accessed on 29 March 2019) and https://dtp.cancer.gov/discovery_development/nci-60/characterization.htm (accessed on 29 March 2019).

### 3.5. Molecular Docking

Docking simulations were carried out using Autodock ver. 4.2 software, while MGL TOOLS 1.5.6 was used to prepare the input files [37,38]. United atom representations were used for all components of the used complexes. Docking runs were performed using a Larmarckian genetic algorithm. Docking began with a population of random ligand conformations in a random orientation and at a random translation. Each docking experiment was derived from 200 different runs, which were set to terminate after a maximum of 5 × 10^6^ energy evaluations or 27 × 10^3^ generations, yielding 200 docked conformations. The population size was set to 300. For all other parameters, default values were used.

All detailed information about the methodologies is given in the Appendix A. Molecular graphics and analyses performed with UCSF Chimera [46,47].

## 4. Conclusions

A series of novel 3,9-disubstituted acridines, **17a**–**17j**, were synthesized with the aim of identifying new potential cytostatic agents. The starting point in the synthetic route was a Jourdan–Ullmann coupling leading to the preparation of *N*-phenyl-substituted anthranilic acid. The design of these substances developed from the intention of studying the influence that the structural variability at position 9 of the acridine scaffold imparts on the biological activity of the derivatives. The entire synthetic plan consists of eight reaction steps that provide the final products in a moderate yield. During the course of the reactions, the formation of side products was also observed, a result which is probably due to the enones, reactive intermediates, which were formed by the elimination of the pyrrolidine ring.

The compounds showed evidence of DNA binding activity (2.81–9.03 × 10^4^ M^−1^), with the results of UV-Vis spectroscopy, DNA melting, and circular dichroism assays indicating that the derivatives act as effective DNA groove binder agents with partial intercalation. The studied compounds were also investigated for their effect on both Topo I and Topo IIα. The results suggest that the ability of the derivatives to inhibit Topo I was reduced when the length of the linker chain between the acridine and the benzene ring was increased. When the inhibition effects of the derivatives on Topo I and Topo IIα are compared, it is apparent that the studied compounds show a preference for Topo I.

Docking simulations were used to determine the possible mode of interactions between Topo I/IIα and the novel disubstituted acridines **17a**–**17j**. The simulations suggest that there is a difference between the way in which acridines **17a**–**17j** interact with Topo I and the way in which they interact with Topo IIα. In the case of Topo I, the acridines directly affect the binding pocket residues, while non-bonded interactions with the nucleotide backbone were more typical for the cleavage complex for topo IIα.

Analyses using cheminformatics and computational chemistry were also used to study the relationship between the physicochemical properties of acridines **17a**–**17j** and their biological activity on a cellular/molecular level. A strong correlation was observed between the DNA affinity of these substances and their lipophilicity. It is possible to conclude that the increase in the DNA binding ability of acridines **17a**–**17j** is driven by an increase in entropy and/or enthalpy that accommodates the formation of intercalation complexes.

The synthetized acridines **17a**–**17j** were also subjected to in vitro screening against a panel of 60 cancer cell lines conducted by the Developmental Therapeutic Program of the National Cancer Institute (NCI). The highest levels of biological activity were recorded for acridines **17a**—aniline, **17b**—*N*,*N*-dimethylaniline, **17e**—benzyl, **17f**—4-methylbenzyl, **17g**—4-fluorobenzyl and **17h**—phenylethyl, with growth inhibition constants, GI50, in the micromolar range. The highest level of biological activity was displayed by aniline acridine **17a** (MCF7–GI_50_ 18.6 nM) and *N*,*N*-dimethylaniline acridine **17b** (SR–GI_50_ 38.0 nM). Anilines **17a** and **17b** only displayed higher cytostatic activity than doxorubicin against the multidrug-resistant ovaria cancer cell line, NCI/ADR-RES.

The relationship between the cytostatic activity of derivatives **17a**, **17b**, **17e**–**17h** and the values of *K*_B_, Log*P*, Δ*S*°, δ were also studied. A significant correlation was only observed in the case of charge density, δ, and it is therefore possible to suggest that the cytostatic effect could be linked to the structural specificity of the acridine derivatives. 

Overall, 3,9-disubstituted acridines **17a**–**17j** appears to be a perspective molecular pattern for the next generation of acridine-based drugs with antiproliferative activity due to their potent inhibitory activity against human Topo I.

## Data Availability

Data are contained within the article and Appendix A.

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
