# Peer review of "Novel 3,9-Disubstituted Acridines with Strong Inhibition Activity against Topoisomerase I: Synthesis, Biological Evaluation and Molecular Docking Study"

_molecules, 2023, doi:10.3390/molecules28031308_

Round 1

Reviewer 1 Report

The article “Novel 3,9-Disubstituted Acridines with Strong Inhibition Activity against Topoisomerase I: Synthesis, Biological Evaluation and Molecular Docking Study” by Krochtová et al. describes eight step synthesis of di substituted acridines at positions 3 and 9. The authors described unwanted reactions during syntheses and prepared compounds with promising anticancer activity. Two compounds are more potent than doxorubicin in the multidrug-resistant ovaria cancer cell line. The article also presents many spectroscopic and theoretical studies including interactions with nucleic acids and enzymes. The article is well written with negligible amount of typos:

Page 5, legend to Table 1, instead of KB should be KB (twice).

Page 9, line 228, Rather than “a negative correlation was observed.”, it should be said “without any apparent correlation.”

Page 9, lines 245-246, Since it was not proved, the part of sentence “although the impact of the d-orbitals on interactions with DNA cannot be excluded” should be omitted.

I will recommend the article for publication after minor revision.

Author Response

Dear reviewer,

We have accepted all of the referees’ recommendations and have made the following corrections to our manuscript. We have responded to the questions one by one, and all of the corrections to the text are highlighted in yellow.

Page 5, legend to Table 1, instead of KB should be KB (twice).

There were corrected the mistakes.

Page 9, line 228, Rather than “a negative correlation was observed.”, it should be said “without any apparent correlation.”

This sentence was corrected.

Page 9, lines 245-246, Since it was not proved, the part of sentence “although the impact of the d-orbitals on interactions with DNA cannot be excluded” should be omitted.

This part of the sentence was deleted.

Reviewer 2 Report

The manuscript is devoted to the investigation of the 3,9-disubstituted acridines with strong inhibition activity against Topoisomerase I. The DNA binding activity and activity against a panel of 60 cancer cell lines also were determined. Docking simulations and cheminformatics results are consistent with the experimental data. The manuscript is well-structured, interesting and contains new data. This work should be improved according to the following comments:

1) In supplementary “Synthesis of N-(9-oxo-9,10-dihydroacridin-3-yl)-3-(pyrrolidin-1-yl)propanamide hydrogen chloride (15)”:

Compound 15 obtained as hydrochloride, but in scheme 1 (L. 79) it was shown as free base. Check for all compounds whether it is a base or a hydrochloride.

2) Clarify in introduction why did you choose the BRACO-19 framework for the synthesis of new 3,9-disubstituted acridines.

3) Your target compounds 17a-17j obtained as crystalline solids. Have you tried growing a single crystal for X-ray diffraction analysis?

4) In 3.1. Chemistry section, there is no description of the NMR spectra for compound 17g.

Author Response

Dear reviewer,

We have accepted all of the referees’ recommendations and have made the following corrections to our manuscript. We have responded to the questions one by one, and all of the corrections to the text are highlighted in yellow.

1) In supplementary “Synthesis of N-(9-oxo-9,10-dihydroacridin-3-yl)-3-(pyrrolidin-1-yl)propanamide hydrogen chloride (15)”:

Compound 15 obtained as hydrochloride, but in scheme 1 (L. 79) it was shown as free base. Check for all compounds whether it is a base or a hydrochloride.

The scheme 1 and 2 were corrected, the structure of the compound 15 is now presented as a  hydrochloride salt.

2) Clarify in introduction why did you choose the BRACO-19 framework for the synthesis of new 3,9-disubstituted acridines.

The last sentence of the introduction part was modified:

Therefore, we decided to prepare new acridine-based disubstituted compounds taking advantage of some structural features of BRACO-19 (6), a molecular pattern with potential anticancer applications.

3) Your target compounds 17a-17j obtained as crystalline solids. Have you tried growing a single crystal for X-ray diffraction analysis?

We did not try to grow a single crystal, we focused mainly on biological aspects of our substances.

4) In 3.1. Chemistry section, there is no description of the NMR spectra for compound 17g.

The missing NMR spectra of the compound 17g was inserted into the manuscript.

Reviewer 3 Report

Manuscript details:
Journal: Molecules
Manuscript ID: molecules-2183718
Type of manuscript: Article
Title: Novel 3,9-Disubstituted Acridines with Strong Inhibition Activity
against Topoisomerase I: Synthesis, Biological Evaluation and Molecular
Docking Study
Authors: Kristína Krochtová, Annamária Halečková, Ladislav Janovec *,
Michaela Blizniaková, Katarína Kušnírová, Maria Kožurková * Submitted
to section: Medicinal Chemistry

Abstract: A series of novel 3,9-disubstituted acridines were synthesized and their biological potential was investigated at a cellular and molecular level using an experimental and a theoretical approach. The selected 3,9-disubstituted acridine derivatives were studied in the presence of DNA, their binding constants were calculated (2.81–9.03 × 104 M-1). A strong correlation between the lipophilicity of the acridine derivatives and their DNA binding ability was identified. The derivatives inhibited topoisomerase I/IIα in a submicromolar concentration. Molecular docking simulations suggested that acridines 17a–17j interacted differently with topoisomerase I and with topoisomerase IIα. Acridines were tested against a NCI panel of 60 cancer cell lines. The strongest biological activity was displayed by N-[9-(phenylamino) acridin-3-yl]-3-(pyrrolidin-1-yl) propanamide (MCF7 – GI50 18.6 nM) and N-(9-{[4 (dimethylamino)phenyl]amino}acridin-3-yl)-3-(pyrrolidin-1-yl)propanamide(SR – GI50 38.0 nM).

Comments

In this manuscript, authors investigated a series of novel 3,9-disubstituted acridines 17a–17j were synthesized with the aim of identifying new potential cytostatic agents. The ways of evaluations are relatively comprehensive, and I like the author stated the purpose of doing each evaluation at the first sentence before get to the detailed result and discuss part. This way helps the reader to follow up each step easily. The manuscript is overall easy to read and the topic is worth investigating as well.

Questions:

1. Since multiple methods and measurements were used in this manuscript and most of them are served for different purposes, the manuscript would be easier to access by adding a chart of the whole workflow at the start. Stating the purpose of each evaluations and the brief result based on certain parameters. A clear workflow would allow the reader to quickly catch up with the whole procedure.

2. The abstract can be more specific and clearer to the main purpose and the main result. At least the whole paper is trying to find potential cytostatic agents should be mentioned.

3. The introduction part provided decent background information to make readers understand what is the significance of doing such an investigation. However, there are some typical cytostatic agent exist and already been using to do similar cancer treatment, such as Anthracyclines or specific medicine like Paclitaxel. Some background of the chemicals which does the same treatment should be mentioned.

4. Following with Q3, those existed chemicals/medicine even needed to be compared in the horizontally with the compounds 17a–17j that the authors were investigating. So that the benefit and the necessity of investigating a series of new agents will be emphasized more.

5. The legends in the manuscript is non-standard. Such as Figure 2 should have labeled as Figure 2 A and Figure 2 B. And Figure 6 have A and B in the description but there no corresponding tabs on the plots